

# Delayed evoked potentials in zebra finch (*Taeniopygia guttata*) under midazolam-butorphanol-isoflurane anesthesia

Pin Huan Yu and Yi-Tse Hsiao

Department of Veterinary Medicine, School of Veterinary Medicine, National Taiwan University, Taipei, Taiwan

Corresponding author
Yi-Tse Hsiao, ythsiao@ntu.edu.tw

## ABSTRACT

Avian animals are visually inclined, which has caused them to attract increasing attention for visual neurophysiology or electrophysiology studies, including the study of the visual evoked potential (VEP). VEP has developed into an investigative tool for understanding the physiology and the pathology of the visual pathway. Chemical restraint is a common method to minimize motion artifacts in animals when acquiring VEP data, but little is known about its influence on the signal in an avian animal. In addition, it is difficult to make comparisons between conscious state data when the animals are ultimately under anesthesia. Therefore, finding drugs and developing protocols that have an acceptable effect is valuable. We compared the local field potentials of physically and chemically restrained zebra finches (*Taeniopygia guttata*), a small avian species, to simulate a relatively challenging recording condition. Finches were sedated with midazolam-butorphanol, and anesthesia was maintained by isoflurane. Electrodes were implanted into the left nucleus rotundus, which is a visual nucleus in birds. The VEPs of the control group (*N* = 3) were recorded after they fully recovered and were restrained by towels. The other birds (*N* = 3) were recorded under anesthesia. The results show that without the visual stimuli, anesthesia generally suppressed the overall power of field potentials. However, by focusing on the spectra during VEPs, visual stimuli still triggered significant VEPs in frequencies below 30.8 Hz, which were even stronger than towel-restrained birds. The drugs also prolonged the latency of the VEP, increased the duration of the VEP when compared to towel-restrained birds. As regard to towel-restrained zebra finches, the field potentials were less synchronized and may need data preprocessing to have clear VEPs. In conclusion, the current study presents evidence of basic VEP for zebra finch under midazolam-butorphanol-isoflurane anesthesia with a protocol that is a safe and feasible anesthetic combination for chemical restraint, which is particularly useful for small animals when obtaining evoked potentials.

## INTRODUCTION

Evoked potentials are voltage fluctuations of nervous systems which are triggered by the occurrences of physical or mental events (*Picton et al., 2000*). Evoked potentials have been

used for intraoperative neurophysiologic monitoring and experimental purposes under anesthesia (*Bithal, 2014*; *Cecchetto, Mahmud & Vassanelli, 2015*). Visual evoked potentials (VEPs) were identified as a neurophysiologic test that could reflect the integrity of the visual pathways. Hence, VEPs came to be used as a tool for intraoperative monitoring during major surgery (*Sharika, Mirela & Dinesh, 2016*). Although the effects of anesthesia on evoked potentials have been studied (*Bithal, 2014*), the presence of anesthetics might be a possible confounding factor and can alter the evoked potentials. It is essential not to alter the pharmacological state of patients to avoid any changes in the recording of evoked responses. It is also important to recognize the responses of evoked potentials under anesthesia during clinical and experimental conditions. However, data describing the effects of anesthesia on VEPs in avian animals under laboratory settings are still insufficient.

Despite anesthetic agents affecting the amplitude and latency of evoked potentials through direct inhibition of synaptic pathways or by indirectly altering the balance between inhibitory and excitatory influences (*Bithal, 2014*; *Sloan & Jäntti, 2008*), evoked potentials obtained from anesthetized monkeys (*Nauhaus et al., 2009*) were still shown to be valuable data compared to those gathered from awake monkeys (*Ray & Maunsell, 2011*). Working under anesthetic conditions is sometimes inevitable, especially when working with animals that cannot be physically restrained. For example, chemical restraint is one of the options to make monkeys watch a monitor for minutes without motion artifacts when studying their visual systems (*Nauhaus et al., 2009*). If the monkeys are not chemically restrained, the monkeys need to be trained over several months in order to record their brain activities of visual systems (*Ray & Maunsell, 2011*). In addition, chemical restraint is helpful in avoiding interfering electromyographic signals (*Brauer et al., 2011*; *Itamoto et al., 2001*) which may generate from eye muscles or movements such as rotation of the eyeballs, blinking, and head turning (*Brauer et al., 2011*). Finally, it can minimize the stress caused by experimental procedures and provide analgesia and autonomic reflex stability (*Dondi et al., 2016*; *Pypendop, Poncelet & Verstegen, 1999*).

Avian animals are visually inclined animals with well-developed color vision that have attracted more attention in avian behavior, ecology, and neuroscience studies (*Martin & Osorio, 2008*). Among all of the avian animals, zebra finch (*Taeniopygia guttata*) is one of avian species that is commonly used as animal models as regard of neuroscience experiments especially in studying the auditory system and language learning (*Brainard & Doupe, 2002*). In regard to visual relative studies, researchers also reported that zebra finches are less interested in potential mates with black beaks (*Burley & Coopersmith, 1987*; *Collins, Hubbard & Houtman, 1994*). At least two visual pathways, the tectofugal and thalamofugal pathway, are identified in avian animals (*Zeigler & Bischof, 1993*). It will be a very intriguing question about how zebra finches use these visual pathways to process color information in their brains (*Bennett et al., 1996*; *Hunt et al., 1997*). Analyzing VEPs from the brain areas of these visual pathways may shed some lights on these questions. In an ideal scenario, performing a surgery that implants electrodes in multiple brain areas

and conducting VEP experiments after the animal is fully recovered from anesthesia could explain the questions best. However, technically, in order to record multiple brain regions of a fully recovered zebra finch, carrying heavy implantation materials including electrodes and connectors is mandatory and may raise ethical concern. To overcome above mentioned limitations, we simplified the procedure by recording VEPs in unanesthetized birds' visual nucleus, rotundus (ROT), which can process color information in tectofugal pathway of birds (*Hodos, 1969*), and compared the VEPs to anesthetized birds that were recorded immediately after implantation. We propose an anesthesia protocol that have acceptable effects on VEPs and can be applied on multiple brain areas recording in zebra finches.

However, using anesthesia in birds is challenging. A study aiming at estimating the risks of anesthetic and sedation-related mortality in companion animals found that the anesthetic/sedation-related mortality in birds ranging from 1.76% to 16.33% depends on different species which was significantly higher than those in dogs (0.17%) and cats (0.24%) (*Brodbelt et al., 2008*). It has been hypothesized that the birds' small size contributes to their higher mortality risk due to higher surface area to volume ratios, high metabolic rates, and trachea opening; the facts described above predispose them to complications like hypothermia, hypoglycemia, and difficulties in maintaining a patent airway. Additionally, the lack of experience of veterinary surgeons with exotic avian species was likely to have contributed to the high perioperative mortality risks (*Brodbelt et al., 2008*). To reduce the mortality rate, balanced anesthetic protocols (*Ilkiw, 1999*), which refers to the use of a mixture of drugs, such that the advantages of small amounts of drugs are used without having to contend with the disadvantages of large doses of any one drug (*Ilkiw, 1999*), are widely used in avian practice to increase the safety of anesthetic procedures (*Kubiak, Roach & Eatwell, 2016*; *Paula et al., 2013*). Butorphanol, an opioid which is a mixed agonist/antagonist that has low activity at μ-receptors and strong agonist activity at κ-receptors, makes it appropriate for avian species. The analgesic effect has been validated in pigeons and psittacines. Midazolam has been used in avian species for sedation. It is administered by intramuscular injection and has been demonstrated to have no significant changes in cardiopulmonary function in Canadian geese (*Branta canadensis*), guinea fowl (*Numida meleagris*), pigeons (*Columba livia*), or quail (*Coturnix japonica*) (*Kubiak, Roach & Eatwell, 2016*). Opioids (Butorphanol) combined with benzodiazepines (Midazolam) are commonly used as premedicants in birds before an inhalational agent such as isoflurane.

The objective of this study was to determine the effects of general anesthesia induced by butorphanol-midazolam-isoflurane on intracranial evoked potentials in zebra finches by comparing it to physically restrained zebra finches. The goal is to highlight the safety and feasibility of this protocol for a relatively small animal under difficult anesthesia conditions, and ultimately to provide a balanced anesthetic option in small experimental targets undergoing VEP recording projects, and to test whether the protocol interferes VEP signals. The current study compares VEPs from conscious and anesthetized states in zebra finches.

## MATERIALS AND METHODS

### Animals

Eight male zebra finches (18–21 g) from a commercial breeder (San-Xing Bird Store, Taipei, Taiwan) were used in these experiments. The animals were individually housed in home cages with temperature controlled at 23 ± 1 °C, and the light-dark cycle (AM 7:00 on, PM 7:00 off) was maintained under a natural photoperiod. Food and water were available ad libitum. Commercial bird feed which contains millet kernel, canary millet, vitamins, minerals (Ho Mei Chien, Taichung, Taiwan) was provided. All procedures performed in this study were approved by the National Taiwan University Animal Care and Use Committee numbered NTU106-EL-00026. Six birds were randomly assigned to one of two groups in which flash-evoked potentials were monitored under physical (Bird 1, 5, 6) or chemical restraint by the midazolam-butorphanol-isoflurane anesthesia (Bird 2, 3, 4). Additional control experiments were performed on two of the eight animals. Bird 7 was sedated by diazepam. Bird 8 was physically restrained but its visual stimuli were blocked by a black ethylene-vinyl acetate (EVA) foam (30 × 25 × 0.5 cm; Yu Yuan Plastic, Taipei, Taiwan).

### Surgery and anesthesia

All birds were oxygenated in an anesthetic chamber for 10 min prior to preanesthetic medication with intramuscular injection of one mg/kg midazolam (Dormicum; Roche Ltd, Fontenay-sous-Bois, France) and one mg/kg butorphanol (Ilium Butorgesic injection; Troy Laboratories Pty. Ltd, Glendenning, Australia) (Hawkins et al., 2018) into the pectoral muscle. A total of 10 min after injection, anesthesia was induced with 3% isoflurane (Attane, Isoflurane; Panion & BF Biotech Inc., Taoyuan, Taiwan) combined with 100% oxygen that was provided by use of a veterinary anesthesia delivery system (A.D.S 2000; ENGLER, Hialeah, FL, USA). The same anesthetic delivery system was used to maintain anesthesia in a handmade face mask to maintain a tight seal without blocking both eyes with 1.5–3% isoflurane adjusted for the anesthetic depth of the bird. The depth of anesthesia was determined through monitoring respiratory rate, corneal reflex, palpebral reflex and wing withdrawal reflex (Heard, 2015).

The skull of the bird was fixed on a stereotaxic frame (Stoelting, Wood Dale, IL, USA) by gently inserting earbars in its external acoustic canals. The coordinates of the target region, the nucleus ROT, were selected based on a stereotaxic atlas of the brain of the zebra finch (AP, 2.6 mm; ML, −2.0 mm; DV, 5.0 mm relative to y point) (Nixdorf-Bergweiler & Bischof, 2007). The tissue over the surface of the skull was removed and a hole was drilled for electrode implantation. Before inserting the electrode, the dura was removed with a 27G needle. A tetrode consisting of four twisted 0.05 mm stainless steel wires (California Fine Wire; Grover Beach, CA, USA) was slowly implanted into the nucleus ROT, which is a visual nucleus that processes color information in birds (Hodos, 1969). Two stainless steel ground screws (M1.0*3 mm) were turned into the right frontal bone. The LFP from the ROT and the potential from right hemisphere were connect to an interface board. For the birds (Birds 2 to 4) that were recorded under anesthesia,

the interface board was linked to a 20X-gain head stage and tethered to the preamplifier of an OmniPlex A system (Plexon, Dallas, TX, USA). Otherwise, the interface board was cemented to the skull with dental acrylic (Tempron; GC Co., Tokyo, Japan), and the subject had recovered for 7–10 days before LFP and VEP recording. Postoperatively, birds in the experimental group were euthanized after completed experimental procedures described in the following section. For the control group, five mg/kg carprofen (Pfizer, New York, NY, USA) was provided by intramuscular injection for 3 days for postoperative analgesia.

## Experimental procedure

The recording was done in a dim (lx: 2.1E) and quiet room (45 dB). We used a towel to physically restrain five of the subjects (Bird 1, 5, 6, 7, 8) and then applied a small amount of Cyanoacrylate (Loctite 415, Henkel, Germany) on the right eyelid, and the subject was placed in front of a laptop (model: SVP132A1CP, SONY, Japan) with the right eye aimed at the screen (Fig. 1, illustrations). The color stimulation slides were made by Microsoft PowerPoint (2016; Microsoft, Washington, DC, USA). In order to further confirm the VEPs were not from non-specific noises such as the static when changing slides, a black high density EVA foam which reflect very little light was used to block the color stimuli from the screen (Bird 8). The foam was lowered and elevated by hands (Fig. S1). For chemically restrained birds (Bird 2 to 4), after the surgical procedure described above, #6-0 surgical sutures were used to open and fix the right eyelids. During the recording, the concentration of isoflurane was turned to 1%. The grounding screw was used as a reference, and the LFP signals were amplified (gain 2,500×), digitized, bandpass-filtered (between 0 and 500 Hz), and recorded (at 2,000 Hz sampling rate) by a 16-channel OmniPlex A system (Plexon, Dallas, TX, USA). All recording was performed in a Faraday cage that was grounded with the amplifier. The laptop screen was approximately 30 cm away from the subject and placed outside the Faraday cage without plugging in the charger. The retinas of avian animals contain four kinds of cone cells and result in their tetrachromatic color vision (*Osorio, Vorobyev & Jones, 1999*; *Viets, Eldred & Johnston, 2016*). The cone cells are sensitive to the wavelengths which are believed to be red, green, blue, and violet to ultraviolet. As a pilot study, we first flashed 20–21 trials of 2 s of light green (RGB color code: (0.25, 1, 0.75), lx: 79E) between 2 s of black (RGB color code: (0, 0, 0), lx: 2.4E) for Bird 1. The same procedures were also done by flashing light red (RGB color code: (1, 0.25, 0), lx: 23E), and light blue (RGB color code: (0, 0.75, 1), lx: 45E) for Bird 1. We selected green as the color stimuli for further experiments since it generated the largest VEPs (see Results section). The time points switching between displaying color or black were saved in the same LFP acquisition file. We also tested the VEPs under the effects of classic sedative agent which commonly used in zebra finch: diazepam (*Wolf et al., 2017*). Bird 7 was used to test the effects diazepam. Four mg/kg diazepam (Astar, Hsinchu, Taiwan) was administrated intramuscularly 10 min before recording. A towel was rolled around its body for restraining (Fig. S2).

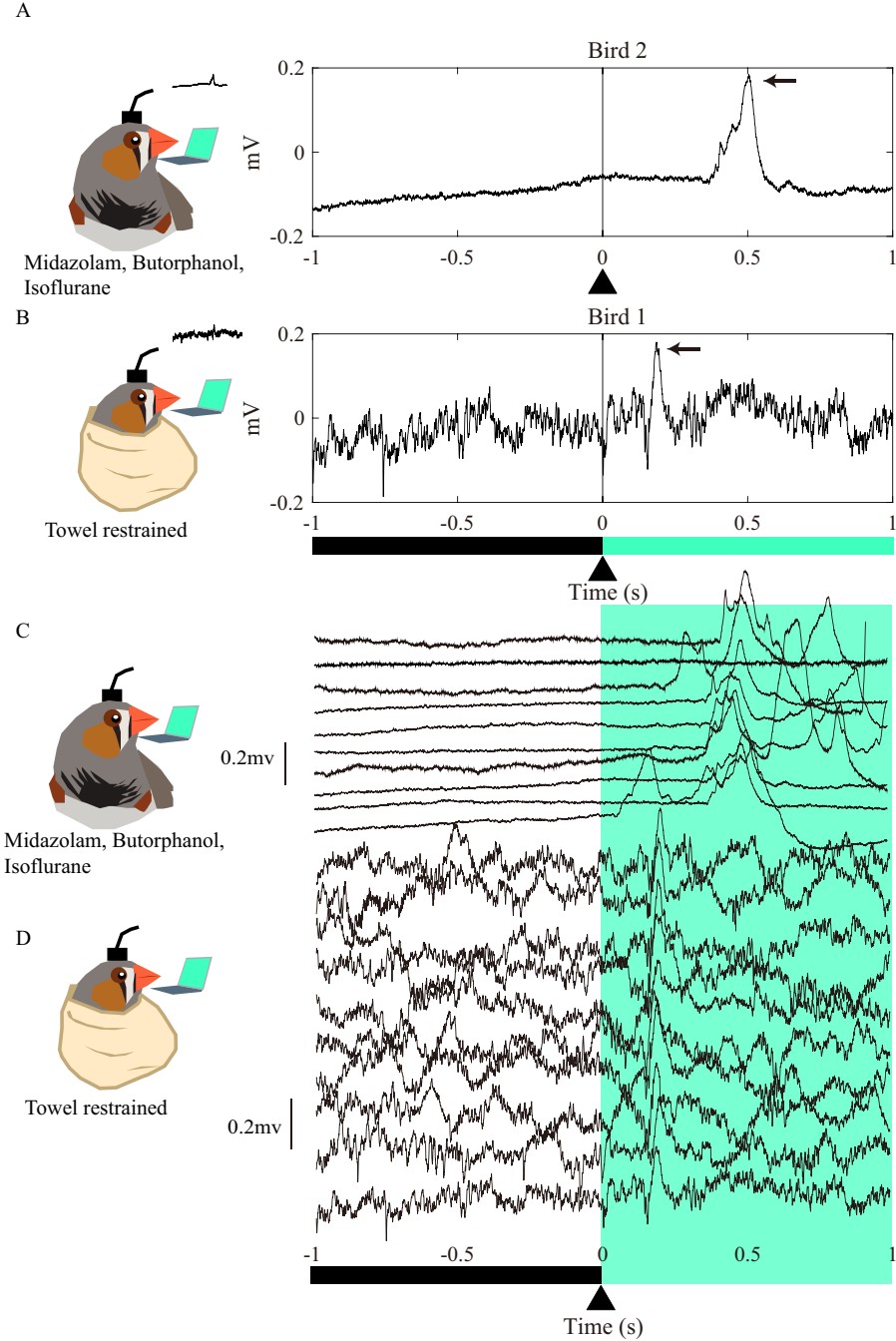

**Figure 1 Examples of raw LFP traces after watching a light under chemical or physical restraint.**
(A) and (B) are graphical abstracts showing the method for recording the evoked potentials.
The zebra finches were either restrained by intramuscular administration of midazolam and butorphanol and maintained by inhalation of isoflurane or rolled with a towel. The electrodes were inserted into the left ROT and recorded the LFP while the right eye watched the screen. The screen displayed blacks and greens. (A) is an example of LFP and the evoked potentials of a chemically restrained bird, (B) was obtained with towel restraint, (C) and (D) are 10 examples of LFP raw traces. The vertical lines and the arrow heads at zeros are the time points when the black switched to green.

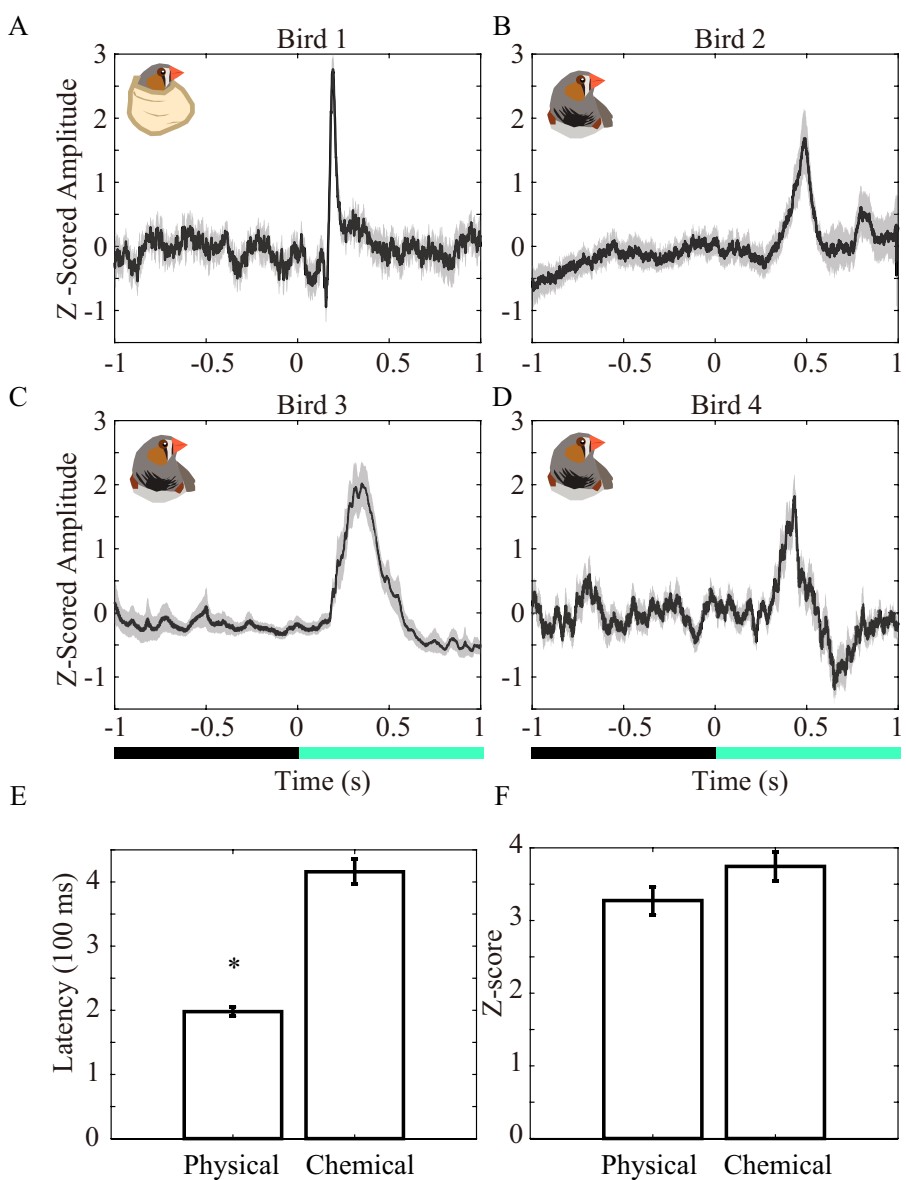

**Figure 2 Averaged LFP traces, peak latencies, and peak amplitudes.** (A–D) Averaged LFP traces for each subject. The illustration in each figure shows the restraint method. The LFPs were normalized by standard deviation and represented with Z-scores. (E) is averaged peak latencies. The Z-scores for peaks are averaged in (F). The lines and shadows and bars and error bars are means ± SEM. Zero second means the time when the screen changed color from black to green. * denotes the $p$-value < 0.01.

## LFP analysis

The analysis scripts were written in MATLAB 2016b (MathWorks, Natick, MA, USA) using the open source toolbox, Chronux (version 2.10) (*Bokil et al., 2010*). We first identified the color or black switch time points and made this time the origin (note as 0 s). Then, the LFP raw traces were cut into ±1 s segments for further analysis. The impedance at the recording site for each bird is difficult to control. Thus, the amplitudes of each ±1-s LFP were Z-scored across time for better comparison between subjects (Figs. 2A–2D). The spectra of

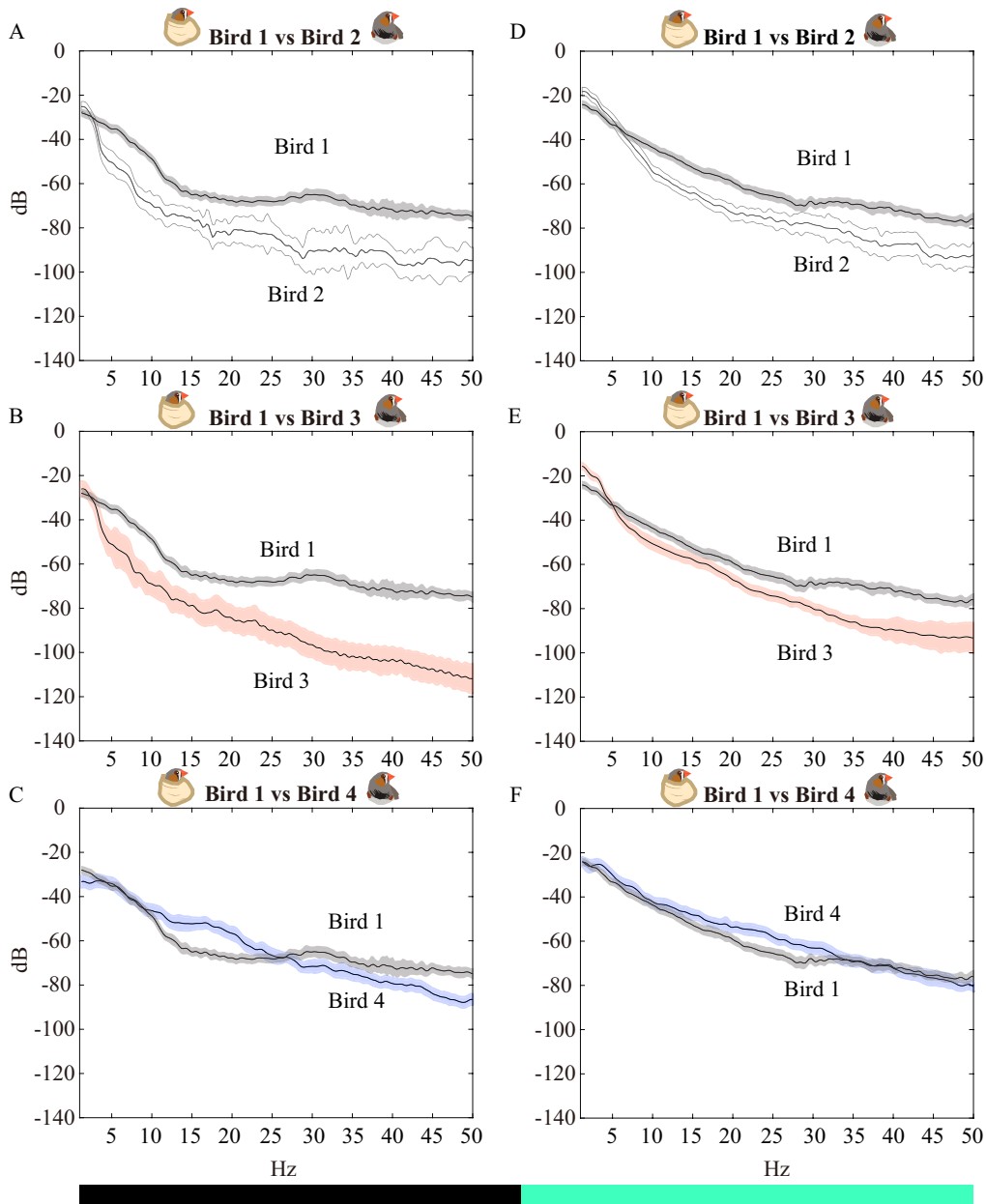

**Figure 3 Pre- and poststimulus spectra.** (A–C) are spectra from 1-s LFPs when watching blacks (e.g., −1 to 0 s), (D–F) are data from 1-s LFP bins after the black changed to green (e.g., 0–1 s). The white, red, and blue bands were means ±95% confidence intervals from Birds 2, 3, and 4, respectively, which were chemically restrained finches. The gray bands are from the physically restrained bird (Bird 1) and illustrate the 95% confidence intervals. For comparison, the spectra of the physically restrained bird were plotted with the chemically restrained birds.          

raw data (spectra of Figs. 3 and 4) were calculated by multitaper spectral analysis, and the error bars were analyzed by the Jackknife method, using the mtspectrumc.m function in Chronux. We set the time-bandwidth product and the number of leading tapers to use to 3 and 5, respectively. The spectrograms (Figs. 4A–4D) were similar to the spectrum but were further investigated in the time domain. We used 0.5-s windows with 0.05-s overlapping

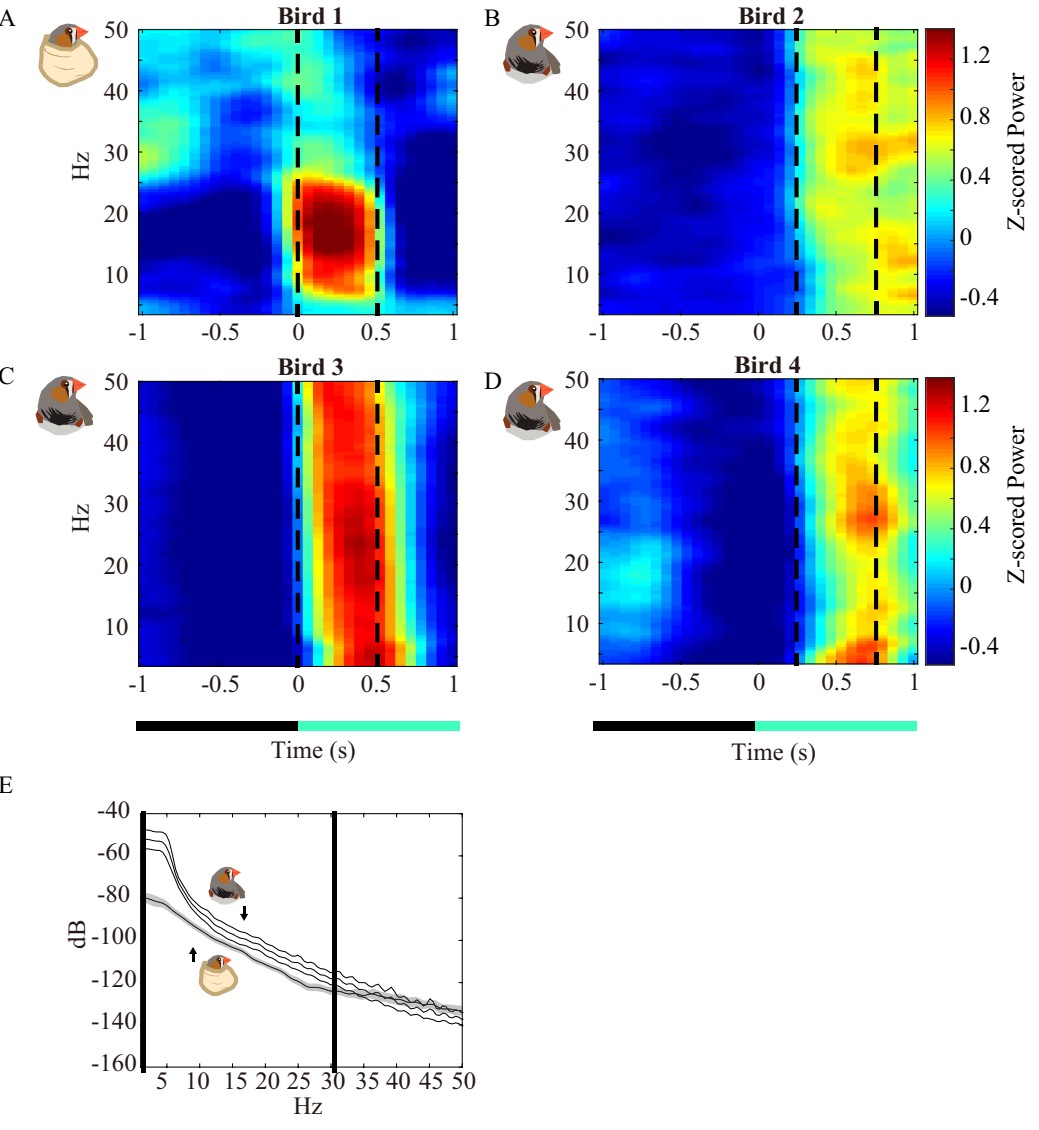

**Figure 4 Averaged spectrogram of each subject and their spectra for VEPs.** (A–D) Averaged spectrogram of each subject. The warmer color indicates stronger Z-scored power at a specific frequency (*y*-axis) during a specific time (*x*-axis). The spectra between the dashed lines are displayed in (E) and are a comparison between bird 1 (shadow) and birds 2–4 (white band). The band or shadow are 95% confidence intervals.

steps, and the time-bandwidth product of the number of tapers are the same as the spectra. The power of the LFP spectrum is correlated with 1/frequency (*Buzsaki, 2006*), which results in low powers of high frequencies. To compensate for this noise, we Z-scored every frequency along the time domain and then Gaussian-smoothed the figure using the boxcar method.

## Exclusion criteria and data preprocessing

Because the response time of VEPs may be interfered by the drugs, using computer programs to exclude data with defined onset period may be not practical. To solved this,

we defined the exclusion criteria by both inspection of the onset of VEPs visually and set an amplitude threshold for including the birds with VEPs. The procedures were done as follows: Each ±1-s raw LFP trace was stacked and plotted (Fig. 1; Figs. S1–S4) for observing the VEPs. The birds with repeatedly showing shooting LFPs after color stimuli were included in the study. To be specific, the delta amplitudes of shooting LFPs were greater than 0.15 mv when compared to the amplitudes at 0 s. The LFPs of excluded birds were further pre-processed with independent component analysis (ICA) for extract the potential data that masked by noises (Figs. S3–S5). An open source tool box, FastICA (*Gävert et al., 2005*) (Available at https://research.ics.aalto.fi/ica/fastica/code/dlcode.shtml), was used for the ICA. Two control birds (Bird 5 and 6) were pre-processed and data were shown in Figs. S3–S5.

## Statistics

Recorded data were evaluated for the quality of the waveforms according to the raw LFP traces. The bird with LFPs that showed enhanced amplitudes to light stimuli were included in the statistics analysis. All results in the figures are depicted as the means ± SEM. The statistical analyses were conducted with SPSS (Version: 10.0.7, IBM, Armonk, NY, USA). For the significant differences of spectra for each frequency (Figs. 3 and 4E), the confidence bands at $p = 0.05$ were calculated by the Jackknife method, using the mtspectrumc.m function in Chronux. This gave the 95% confidence intervals of the spectra (*Bokil et al., 2010*). Detailed methods are described in the Results section. We used one-way ANOVA to analyze significant differences for the latencies and peak amplitudes. A level of $p < 0.05$ was taken as indicating a statistically significant difference.

## RESULTS

### The drugs prolong the latency of VEP

Electrodes were implanted into the ROT in eight zebra finches. In the control group, the electrodes were glued to the birds' skulls, and the LFPs were recorded after 7–10 days of convalescence. However, only 1 bird (Bird 1) showed waveforms that met the preset criteria (Fig. 1; Figs. S3–S4) for further analysis. In the experimental group, VEPs were recorded during the implantation surgery under anesthesia. The raw LFP traces of anesthetized birds (Figs. 1A and 1C) showed a predominance of low frequencies, which represented smoother LFPs (Figs. 1A vs 1B, before the light stimulation). Then, the visual pathway was further stimulated with light green (color bar at the bottom of Fig. 1B) because we found that zebra finches are sensitive to this color (z-scored VEP peak amplitudes for green: 2.8 ± 0.2, blue: 2.2 ± 0.2, red: 0.8 ± 0.2 in Bird 1). The examples revealed that the latency of VEPs was longer in the experimental group than in towel-restrained birds (Arrows, Figs. 1A vs 1B).

We further analyzed the LFPs of every trial. The impedances for each bird were not the same, and they consequently affected the amplitude of the LFP. Therefore, we z-scored the LFPs and averaged across every stimulus (Figs. 2A and 2D). The evoked potentials spiked at 415.9 ± 20.8 ms, which was significantly longer compared to control birds (Fig. 2E, 197.9 ± 7.3 ms, $F_{(1,80)} = 40.8$, $p < 0.01$). In contrast, the amplitudes of evoked

potentials were similar between the chemically (3.7 ± 0.2) and physically (3.3 ± 0.2) restrained birds (Fig. 2F, $F_{(1,80)}$ = 1.7, $p$ = 0.19). This finding suggested that acute recording (recording right after implantation) was not different from delayed recording in terms of the amplitudes of VEPs. Similar to Fig. 1, the duration of VEPs was also longer in chemically restrained birds than in physically restrained birds (Figs. 2A vs 2B–2D). We further salvaged the control data which have weak VEPs when inspected by raw LFP traces (B5; Fig. S3 blue traces, B6; Fig. S4 blue traces). An ICA pre-processing was applied on the raw LFPs of Bird 5 and Bird 6 (B5; Fig. S3 red traces, B6; Fig. S4 red traces). Although the VEPs were not obvious, their z-scored traces, spectrograms, and ICA processed spectrograms still represent VEP signals (Fig. S5). The z-scored amplitude were 1.3 ± 0.3 at 190 ms in Bird 5 and 1.7 ± 0.2 at 216 ms in Bird 6 (Figs. S5A and S5D).

We were also curious about whether the VEPs were from some nonspecific artifacts which generated by switching between the color and black slides in the screen. Thus, we used a black high density EVA foam to physically block the green light from the screen and conduct a towel restrained experiment again (Bird 8; Fig. S1). The VEPs were still generated (Figs. S1A and S1B) although the waveforms are wider than Bird 1 (Fig. 2A). This phenomenon can be explained by the speed limitation for lowing and elevating the foam by hands when compared with the snap changes of slides by PowerPoint.

## The drugs decreased overall power but not VEPs

For further inspecting which bands were mainly affected, we cut the raw LFP traces into ± 1-s segments starting from the flash time points, which represent zero (see Materials and Methods: *LFP Analysis*). Then, the LFPs were transformed into pre- and poststimulus spectra (prestimulus: Figs. 3A–3C; poststimulus: Figs. 3D–3E). The 95% confidence intervals of the spectra were showed by the shadows and bands. Figure 3 displays that the frequencies >5 Hz were diminished by the drugs, but frequencies <5 Hz were unaffected, which is consistent with the results observed from LFP traces (Figs. 1 and 2A–2D). These spectra suggested that the local computations of neurons that oscillate at high frequency were abolished by the drugs (*Buzsaki & Schomburg, 2015*; *Zheng et al., 2016*), but the ROT still generated the brain waves that synchronized with other brain regions. When inspecting the 1-s spectra after stimuli (Figs. 3D–3F), we noticed that changes (Δ power) between the physically and chemically restrained birds were decreased. Thus, we analyzed the time domain of the spectrum to illustrate the spectrograms (Figs. 4A–4D). The dynamics of the spectra echoed the results in Figs. 3A–3C, which showed relatively weak power before light stimulation compared to the physically restrained bird. However, the power of VEPs were not diminished by drugs (Figs. 4C–4D between dashed lines). Therefore, we cut the LFP traces between the VEPs (Figs. 4A–4D between dashed lines) and calculated their spectra (Fig. 4E). Interestingly, by comparing the confidence interval of the spectra, the frequencies below 30.8 Hz (Fig. 4E between solid lines) were significantly stronger in chemically restrained finches. The data implied that although midazolam-butorphanol-isoflurane generally suppressed the functions of local networks (*Buzsaki & Schomburg, 2015*; *Zheng et al., 2016*), light cues still triggered strong evoked potentials (Fig. 4E), suggesting that the anesthesia protocol is feasible for recording evoked potentials.

In addition to testing this three-agent anesthesia protocol for further multi-regional intracranial recording, we were also interested in comparing the influences of the traditional sedative, diazepam. Diazepam increases the effects of the inhibitory neurotransmitter gamma-aminobutyric acid (GABA) (*Riss et al., 2008*) and can be used as a sedative for zebra finches in order to record activities of auditory systems (*Wolf et al., 2017*). Diazepam (four mg/kg) provides only a sedative effect instead of an anesthetic effect (*Heard, 2015*) so it can not be used alone during an acute implantation recording. For this reason, we test the effects of diazepam in a towel restrained zebra finch. Figure S2 shows that four mg/kg diazepam did not suppress the amplitudes of LFPs when compared to general anesthesia (Figs. 1, 2 and 4; Bird 2–4). Moreover, the latency of VEP was not interfered by four mg/kg diazepam (peaked at 174.5 ms; Fig. S2A).

## DISCUSSION

### Designation of the present anesthetic protocol

The clinical usefulness of flash VEPs is well known in human medicine because it allows objective assessment of the functional integrity of the visual pathways from the retina to the visual cortex, even during general anesthesia or coma conditions (*Dondi et al., 2016*). However, a few decades ago, VEPs could not be reliably interpreted intraoperatively, and lack of consistency made VEPs a less effective monitoring tool than other modalities during anesthesia, which suggested variable influence on the potential by anesthetic agents (*Cedzich & Schramm, 1990*). Therefore, successful evoked potential monitoring requires an adequate understanding of how anesthetic drugs and physiological variations affect signals and how to improve the sensitivity of neuromonitoring through appropriate drug selection and administration (*Soghomonyan et al., 2014*). Evoked potentials are highly sensitive to fluctuations in physiological parameters, such as peripheral and core body temperature, arterial blood pressure, hematocrit, and others (*Soghomonyan et al., 2014*). VEPs are the most sensitive sensory evoked potential to anesthetics in general because the evoked responses travel via polysynaptic pathways (*Kumar, Bhattacharya & Makhija, 2000*; *Soghomonyan et al., 2014*). In general, inhalational anesthetics are more potent suppressors of evoked potentials than intravenous agents. Therefore, balanced general anesthesia with low doses of inhalational agents combined with injectable agents may be recommended (*Soghomonyan et al., 2014*).

To reduce the dose of inhalational anesthetics and therefore avoid its' suppression effects on evoked potentials, a balanced anesthetic protocol was selected in the current study to provide a stable physiological response in anesthetized avian patients and create a less sensitive anesthetic and proper analgesic combination for VEP monitoring. Opioids are known to alter sensitive evoked potentials minimally compared to inhalation agents (*Soghomonyan et al., 2014*). *Schwender et al. (1993)* demonstrated that opioids produce powerful analgesia and have little or no effect on middle latency auditory-evoked potentials (MLAEP). In humans, midazolam at induction doses (0.2 mg/kg) in the absence of any other agent results in slight depression of cortical somatosensory evoked potential amplitude and has minimal effect on subcortical and peripheral components. However, adding an opioid to midazolam preserves cortical somatosensory evoked

potentials better compared with adding opioids to thiopentone or etomidate (*Bithal, 2014*). Combining isoflurane increases flexibility and safety in the protocol as anesthetic depth can be adjusted instantaneously according to the individual's physiologic response during the procedure (*Heard, 2015*). In prior research, the midazolam-butorphanol combination demonstrated an isoflurane-sparing effect in birds, which minimizes side effects caused by isoflurane during anesthesia and during VEP monitoring (*Curro, 1994*; *Curro, Brunson & Paul-Murphy, 1994*). Therefore, a balanced anesthetic protocol was finally designed considering both the safety of the avian animals and the representativeness of the evoked potentials.

## Effects of anesthetic combination

Few studies have focused on the effect of anesthetics on flash VEPs in humans (*Kumar, Bhattacharya & Makhija, 2000*) or animals. A pilot study conducted in birds of prey showed that both the peak latency and wave morphology from normal animals obtained solely under isoflurane anesthesia were similar to those obtained previously in other animal species (*Dondi et al., 2016*). This test can be easily and safely performed in a clinical setting in birds of prey and could be useful for an objective assessment of visual function (*Dondi et al., 2016*). However, there was no control group to describe the effects of isoflurane on the VEPs of birds of prey.

The increased latency in our current study is comparable to the finding that administration of midazolam-butorphanol tended to increase the latency of MLAEP due to its' induction of profound neuroleptanalgesia (*Pypendop, Poncelet & Verstegen, 1999*). Furthermore, isoflurane tends to increase the latency and decrease the amplitudes of VEPs (*Kumar, Bhattacharya & Makhija, 2000*). We speculated that the threshold of action potentials may be increased by anesthetics (*Ries & Puil, 1999*), resulting in difficulty with fast responses (*Brown, Lydic & Schiff, 2010*), and finally delay the VEPs. However, the power of the VEPs below 30.8 Hz were not diminished but were enhanced. If we observe the LFP without the VEPs, suppression of high frequency field potentials (Figs. 2A–2D and 4A–4D) is common in anesthetized subjects (*Hagihira et al., 2002*; *Purdon et al., 2013*) or animals in deep sleep stages (*Murphy et al., 2011*). Reports shows that frequencies above 10 Hz are suppressed by anesthetics (*Hagihira et al., 2002*; *Purdon et al., 2013*). We demonstrated that the VEPs with frequencies below 30.8 Hz were more prominent after chemical restraint (Fig. 4E). We think it is because the sedatives and anesthetics improve the signal-to-noise ratio (*Banoub, Tetzlaff & Schubert, 2003*; *Brauer et al., 2011*; *Sloan, 1994*) and the VEPs from subcortical recording method is less sensitive to anesthetics (*Banoub, Tetzlaff & Schubert, 2003*). Firstly, the sedatives and anesthetics reduced movement artifacts or other electromyography artifacts (they usually introduce noise at higher frequencies) that normally mask the evoked potentials of physical restrained animals (*Banoub, Tetzlaff & Schubert, 2003*; *Brauer et al., 2011*; *Sloan, 1994*). Secondly, subcortical activities are less sensitive to anesthetics when compared to cortical activities because the cortical activities are usually involved in polysynaptic pathways (*Banoub, Tetzlaff & Schubert, 2003*), and the anesthetics predominantly act on synaptic

transmission (*Banoub, Tetzlaff & Schubert, 2003*; *Richards, 1983*). Combining the two advantages made VEPs more prominent and easier to observe.

## Alternative anesthetic options

Recent studies have employed TIVA during VEP monitoring. The apparent improvement in recording VEPs intraoperatively when switching from inhalational anesthesia to TIVA led investigators to examine the role of other anesthetic agents (*Sharika, Mirela & Dinesh, 2016*). Among these, dexmedetomidine, an α-2 agonist, which differs from other anesthetics by its non-GABA mechanisms of sedation and anxiolysis, was studied as an adjunct to TIVA. If baseline VEPs can be obtained in a patient, then dexmedetomidine does not interfere with the acquisition of intraoperative VEPs (*Rozet et al., 2015*). Although TIVA improved the chances of recording intraoperative VEPs reliably, it is impossible to administer TIVA in zebra finches due to their small size. An intraosseous route with a very delicately calibrated syringe pump might be an anesthetic option in the future (*Briscoe & Syring, 2004*). An unignorable fact is the profound cardiovascular side effects (*Hornak et al., 2015*). The safety of using dexmedetomidine in the already challenging avian anesthesia should be considered, especially when the dose range of dexmedetomidine for finches is not known currently (*Hawkins et al., 2018*).

## Comparison of waveform performance

In a previous study in bird of prey (*Dondi et al., 2016*), the waveforms of VEPs were complex since they used electroencephalography by applying electrodes under the skin of the skull, which obtain the signals from relatively large areas. In contrast, we recorded LFPs from the ROT, which receives signals from the optic nerve and is the first relay area of the visual pathway. By analyzing VEPs in this area, the data should be more precise to reflect the effect of drugs when compared with physically restrained subjects, for which it is difficult to eliminate artifacts from muscles when using skull electroencephalography. Despite this issue, a nonsedated brain may inhibit the ROT and give weaker VEPs (Fig. 4E).

## Limitations and future work

The limitations of the current study included that an ophthalmic examination and neurologic exam was not conducted prior to the experiment due to the small size of the finch. However, there was no obvious ophthalmic defect or visual impairment observed before starting the experiment. There has been little discussion in the literature of anesthetic effects on purely VEPs in birds. Therefore, our anesthetic protocol was created based on reports of either sensory evoked potentials or MLAEP. A similar situation was found again when discussing anesthetic effects in the current study. The proposed effects of anesthetics including increased peak latencies, suppression of high frequency waves were supported mainly based on those concluded in existing reports of mammals (*Kumar, Bhattacharya & Makhija, 2000*; *Nauhaus et al., 2009*). It should be further verified in avian animals and with a well-controlled study in the future.

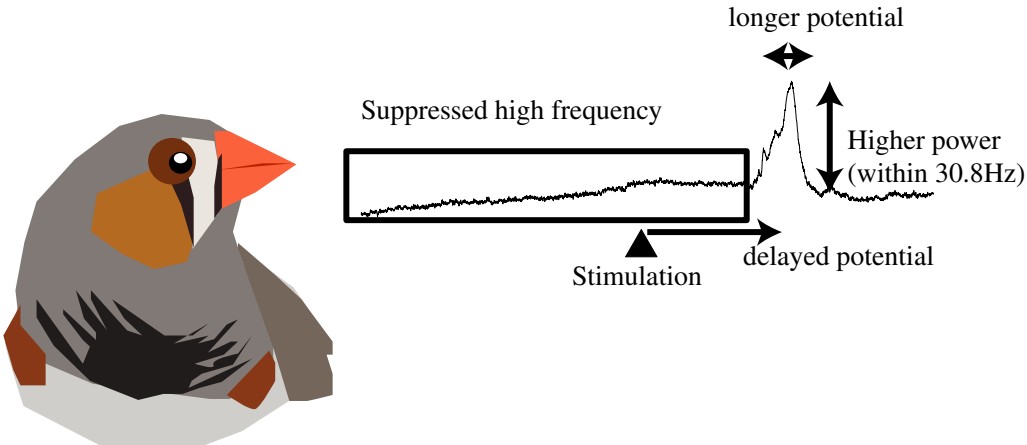

Midazolam and Butorphanol and Isoflurane

**Figure 5 Graphical summary of the influence of midazolam-butorphanol-isoflurane on LFP.** Although the drugs affected the LFP, the protocol is still feasible for testing responses of the visual pathway with VEPs.

## CONCLUSIONS

In the current study, zebra finches were given midazolam-butorphanol-isoflurane and had VEPs recorded to determine a balanced anesthetic protocol. Our results demonstrate that flash VEP could be recorded under the protocol because VEP power below 30.8 Hz did not weaken with chemical restraint (Fig. 5). On the other hand, our results showed that physical restraint in small animals was not ideal due to the data from physically restrained birds being difficult to acquire as a result of motion artifacts and the attention of the bird. We repeated the procedure in three birds, but only one bird showed VEPs. Finally, the lower power of VEPs (below 30.8 Hz) of the physically restrained bird (Fig. 4E) may be caused by the summation of voltage fluctuations of several active brain areas which result in a less synchronized status (*Ray & Maunsell, 2011*). As mentioned previously, LFP data contains more information when recorded in several brain regions simultaneously but it is not as easy to accomplish in zebra finches owing to their high mortality rate during surgery. Although there was minor interference of the received signal (delayed evoked potentials) with the current anesthetic protocol, the benefits (such as application to multi-regional recording in avian animals) were enough to compensate for those drawbacks. Therefore, it is recommended that VEPs be recorded in small animals under anesthesia, and the anesthetic protocol and basic VEP waveform of the zebra finch under the midazolam-butorphanol-isoflurane anesthesia presented in the current study can be applied for similar electrophysiological studies and clinical applications in other species.

## ACKNOWLEDGEMENTS

The authors thank National Taiwan University Veterinary Hospital for technical support of the study.

### Funding

This study was supported by grants from the Ministry of Science and Technology (MOST107-2311-B-002-005 and MOST105-2911-I-002-520). The funders had no role in study design, data collection and analysis, decision to publish, or preparation of the manuscript.

### Grant Disclosures

The following grant information was disclosed by the authors:
Ministry of Science and Technology: MOST107-2311-B-002-005 and MOST105-2911-I-002-520.

### Competing Interests

The authors declare that they have no competing interests.

### Author Contributions

- Pin Huan Yu conceived and designed the experiments, performed the experiments, analyzed the data, contributed reagents/materials/analysis tools, authored or reviewed drafts of the paper, approved the final draft.
- Yi-Tse Hsiao conceived and designed the experiments, performed the experiments, analyzed the data, contributed reagents/materials/analysis tools, prepared figures and/or tables, authored or reviewed drafts of the paper, approved the final draft.

### Animal Ethics

The following information was supplied relating to ethical approvals (i.e., approving body and any reference numbers):

All procedures performed in this study were approved by the National Taiwan University Animal Care and Use Committee numbered NTU106-EL-00026.

### Data Availability

The raw LFP data and MATLAB code are available as Supplemental Files.

### Supplemental Information

Supplemental information for this article can be found online at http://dx.doi.org/10.7717/peerj.7937#supplemental-information.

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
