# Peer review of "Delayed evoked potentials in zebra finch (Taeniopygia guttata) under midazolam-butorphanol-isoflurane anesthesia"

_PeerJ, doi:10.7717/peerj.7937_

## Round 0.1 · original submission · Major Revisions

Your manuscript has now been seen by two reviewers. You will see from their comments below that while they find your work of interest, some major points are raised. We are interested in the possibility of publishing your study, but would like to consider your response to these concerns in the form of a revised manuscript before we make a final decision on publication. We therefore invite you to revise and resubmit your manuscript, taking into account the points raised. Please highlight all changes in the manuscript text file.

·

Basic reporting

The text uses professional and technical language appropriate for a publication. However, there are sentences which are vague or unclear and require clarification, and a couple of sentences would be better re-written in correct English. I think improving the clarity of the manuscript is a critical factor for acceptance to the journal. To assist the authors with this, I have provided specific line comments for the abstract:
[Line 18] A more correct sentence would be "including the study of the visual evoked potential" However, if it had been plural the following would be correct "including the study of visual evoked potentials". This is a minor issue, but best addressed in order to meet the publishing standards of the PeerJ journal.
[Line 21-22] “Controlling the depth of anesthesia and obtaining correct data that resembles a conscious state are contradictory goals.”
Although I believe I understand the intended meaning, the wording is vague and could be clearer. For example, what constitutes 'correct' data? Correct for what purpose? Is the intention to highlight the difficulty in making conclusions about conscious state data when the animals are ultimately under a dose of anaesthetic (a problem of making conclusions about the brain state)? Or is the purpose to highlight the practical difficulties in finding a level of anaesthetic which simultaneously maintains a level of conscious awareness sufficient to complete a given task? Can the authors please state more clearly what is meant by "correct data" and "contradictory goals". What is "correct" and what is the "goal" are the specifics which are not clear.
[Line 30-31] “Although anesthesia generally suppressed the overall power of field potentials, visual stimuli still triggered significant VEPs,”
What frequency were the stronger VEPs at? It was previously stated that the overall power of the evoked potentials decreased, hence it is not clear what frequencies are stronger and what the comparison is. Is it between the anaesthetised VEPs and the towel-restrained VEPs?
[Line 32] “the current study provides basic VEPs” An alternate phrasing could be "provides evidence of basic VEPs" or "presents evidence of basic VEPs". "provides basic VEP waveforms" does not make a lot of sense.
It is also not clear to me in the abstract what the conclusions about the towel-restrained animals are, compared to the chemical restraint.
Next, for the introduction, there are a few vague sections:
[Line 41-43] “, it is essential not to alter the pharmacological state of the patient to avoid any changes in the recording of evoked responses and to recognize the response under anesthesia during clinical and experimental conditions.”
If the authors could break up this sentence and try to make the meaning clearer, it would help with understanding the manuscript. My interpretation is the authors wish to make a statement about how the presence of anaesthetics can alter the evoked potential and might be a possible confound with any given experiment, but I think this could be stated more explicitly and concisely by breaking up the sentence into smaller ones. By “recognize the response” do the authors mean “it is essential to identify the response under anesthesia”?
[Line 70] “difficult anesthesia conditions”
I think the logic behind 'difficult anesthesia conditions' is missing - what is the benefit of doing something so difficult? Knowing more about the potential benefits of studying the zebra finch would help strengthen the argument that this is a worthwhile experiment. Furthermore, I wonder what the challenges are of this method. Is it because anaesthesia on birds is difficult that the authors wish to find better ways of doing it? Does the size of the bird matter significantly in doing experiments? (For example, is it easier to restrain a smaller bird than a larger one?)
A later example of unclear language:
[Line 71] “provide another anesthetic option in small experimental targets undergoing similar projects”
What kind of projects? What is 'similar' about them, and what are the 'current' anesthetic options available? What makes them insufficient? So far we have just learned about one method of anaesthesia - the one that will be applied in the paper. Is that the only possible way of doing anesthesia in a zebra finch?

In terms of literature references and background information, in the vast majority of cases, appropriate citations are included. Where marked, some statements are missing citations and reference to the literature. I think more citations could be included for sections where more background information is required. I think this is tied to the importance for improving the clarity of the manuscript. For example:
[Line 36] “Evoked potentials” (first line of introduction) I think a definition of an evoked potential at the beginning would make it clear what the object of study is for a naive reader.
[Line 40-41] “Although the effects of anesthesia on evoked potentials have not been thoroughly studied”
Actually, there have been a lot of papers about the effects of general anaesthesics on the evoked response. My question is what specific aspect of the literature do the authors believe is lacking, and how does this paper try to address that?
[Line 46-48] “evoked potentials obtained from anesthetized monkey were still shown to be valuable data compared to those gathered from awake monkeys (Nauhaus et al. 2009; Ray & Maunsell 2011).”
This sentence seems to be an important part of the argument the manuscript is making, and yet the specifics of the experimental findings are not elaborated on. Why were some monkeys anaesthetised and others waking and what valuable information could be gathered from the anaesthetised animals that couldn't be obtained from the waking animals?
[Line 50-51] Unwanted effects of blinking, head turning, and motion artifacts would be most important for electrophysiological experiments, and yet this particular detail is not mentioned in the sentence so it is not necessarily clear to the reader.
[Line 54-56]” Avian animals are visually inclined animals with well-developed color vision that have attracted more attention in avian behavior, ecology, and neuroscience studies”
I am naive to neuroscientific studies in avian animals. As a naive reader, I would like to the authors to elaborate upon this point by listing a few examples of the usefulness of birds as model organisms.
[Line 56] Punctuation error after the citation ending in “2008”.
[Line 56] “using anesthesia in birds is challenging”
What makes using anaesthesia in birds challenging? Is it difficult to administer the anaesthetic? Please elaborate so the readers can understand the difficulties involved in this experimental method.
[Line 56] “balanced anesthestic protocols”
Can the authors please provide a brief definition of 'balanced' anesthetic protocol? As mentioned later, they arereferring to the use of multiple anaesthetic drugs during the procedure, though this is not mentioned until later.
[Line 58-59] “μ-receptors and strong agonist activity at κ- receptors”
These receptor types are specific for opioid drugs, and yet the fact that butorphanol is an opioid is not revealed until later in the sentence. If would be clearer if it was mentioned earlier.
[61-66] This section is very clear and makes good use of examples with multiple avian species.
[Line 68] “intracranial evoked potentials in zebra finches by”
One piece of information that is missing in the manuscript so far is why the authors have chosen the zebra finch as the animal model of study. As a reader I would be interested in learning about what kind of studies the zebra finch have been used in, and how the authors are progressing the field by improving or adding to the methodology of said field.
[Line 72] “sedation protocol could potentially benefit small animals”
What is the main benefit expected from using this protocol? Please be explicit.

In terms of the article structure, figures, tables, and raw data; the article is well-organized and structured. The figures are clear and easy to understand. Although the data of the animals included in the paper are included in the supplementary materials, those animals excluded from the analysis has not been included. This makes it difficult to make a judgement as to whether their exclusion was appropriate or not, particularly since the reason for their exclusion is not always stated clearly throughout the paper. One explanation was due to the lack of a visual response, the other was due to motion artefacts. Both are possible, given the difficulty of electrophysiological recordings, but a clearer statement of this would be appreciated. On that note, I wonder why the excluded animals must be excluded for all analyses? As the visual stimulus was periodic (a regular flash stimulus), does using the chronux power spectrum function over all 20 trials instead of on a per-stimulus basis (1-second chunks) allow the frequency of the stimulus to be analysed? This could mean some of the excluded data could still be used for parts of the paper. Alternatively, I wonder if the authors had tried methods of filtering the noise or artefacts? For example, by restricting the frequency analysis around the stimulus frequency or by using a technique like independent components analysis (ICA). In some cases noise can ruin a recording, but there is no mention of attempts to fix this issue, so as a reader, I cannot tell if what the authors have done (or perhaps not mentioned) is sufficient.
I think if more information about the statistics used (e.g. the frequency band analysis) and dealing with the control animals (e.g. exclusion criteria) were included in the manuscript, it could then be acceptable for publication.
The authors have written a very self-contained paper. However, the hypotheses are not stated clearly and some sections of the paper are confusing as mentioned earlier. If the comments mentioned thus far could be addressed, I think the hypotheses and goals of the manuscript would be clearer and the manuscript would be more effective.

Experimental design

The authors have presented original research which fall under the category of biological sciences, which is within the scope of PeerJ.
The research question is unclear. I think it is clear that the gap to be filled is knowledge about how to conduct electrophysiological recordings with zebra finch. However, what is not clear is why this is necessary, and how this information from the zebra finch can be useful. I believe it probably can be useful, but the authors have not made it explicit what sorts of research questions concerning VEPs and zebra finch specifically is interesting (colour vision is mentioned at some point, it would be interesting to read more about that). Information about this is especially lacking in the introduction. This ties into the ‘clarity and background information’ critique earlier.
The authors received ethics clearance for their experiment (specifically, approval by the National Taiwan University Animal Care and Use Committee; NTU106-EL-00026). It is also mentioned in the acknowledgements that the study had technical support from the National Taiwan University Veterinary Hospital. In addition to this, the reader can see that the authors have used a minimal number of animals, and attempt to reduce physical discomfort and psychological distress in the animals with various drugs and a calming environment. The authors describe their experimental procedure clearly, and demonstrate a high level of technical expertise by using precise anatomical terminology and naming equipment used. I do not think anything needs to be changed about the protocol and experiment in order for the manuscript to be accepted. I think what is most necessary is the authors try help the readers understand what the experiment was and why it was done.
On this note, for the most part, the methods are described in sufficient detail. However, I have made specific notes where more information would be beneficial to help the reader understand the experiment and aid in replication. I include some specific examples and comments from the materials and methods section:
[Line 77] “commercial breeder”
Who or what company is the breeder?
[Line 79] “light-dark cycle”
What hours were the lights switched on and off?
[Line 79] “Food and water”
What kind of food do they eat and what is the nutritional composition?
[Line 95-96] “The depth of anesthesia was determined through monitoring respiratory rate, corneal reflex, palpebral reflex and wing withdrawal reflex”
Is there a citation the authors can add which describes these methods of measurement as they relate to birds?
[Line 104-105] “which is a visual nucleus that processes color information in birds (Hodos 1969).”
I think this is a good addition of information and is an example of the ways more background can be added to other parts of the paper.
[Line 112] “continuing”
Completed?
[Line 113] “carpofen”
This is an anti-inflammatory drug, as I understand it. If so, why was this only administered to the control group if it might help the anesthesia group with recovering from the surgery? My understanding from some quick outside reading is that carpofen can interact with multiple kinds of drugs and can be dangerous in some contexts. Do the authors think this might be a confound in the experiment? What is the reason for using this drug as opposed to some other option?
[Line 117] “The recording was done in a dim and quiet room.”
Is there a LUX measurement for the room the authors can report so other experimenters can replicate the luminance? I see they have provided LUX measurements for other aspects of the experiment, namely the stimulus, which is useful, but a few extra details would help, too.

Validity of the findings

The authors have implemented a new method for recording visual evoked potentials from the zebra finch, which is a valid experimental procedure.
The raw data is electrophysiological data and it appears the analyses are appropriate. Just from reading, the statistics are sound, though it would be useful to be able to double-check with some SPSS code considering we have other aspects of the code (though I am not sure how much code is required to be included for supplementary materials). I do not think the lack of extra code is a main concern in the paper’s acceptance.
The experiment has a control group, which is appropriate. However, some animals were excluded due to motion artefact or a lack of visual response. It would be nice (though not necessary) if that excluded data were presented, even as a supplement, since there are so few control animals in the dataset. What is important to note is that it is not clear whether the animals had no visual response or whether they were not detectable due to noise. As long as the reason for their exclusion is clear, then I think the readers can accept the reason why there is only one remaining control animal. However, without that justification, it is difficult to accept an N of 1 for a control group. As mentioned, I wonder if the authors have tried to pre-process the data so it is possible to present the data for the other control animals? I think if the authors reassure the reader they have done all they can to salvage the control data then the manuscript will be in a better position for acceptance.
I include specific comments about the statistics section here:
[Line 152-153] “The bird with LFPs that showed enhanced amplitudes to light stimuli were included in the statistics analysis.”
The word “enhanced” is a bit vague here. Enhanced compared to what? Was there a z-score threshold used to determine whether to include the birds or not? Or was there a check so that those animals which showed a light response were included while those which didn’t were excluded (a binary judgement)? Please provide more detail.
[Line 161] “preset criteria”
Please be clear about what the criteria were for including or excluding animals.
[Line 162-163] “In the experimental group, VEPs were recorded during the implantation surgery under anesthesia.”
Ah, is this the reason the control group received an anti inflammatory after surgery and the experiment group didn't? Hence, the experiment group had VEPs recorded during surgery, while the control group had a recovery time before surgery. Do the authors think this might be an experimental confound?
[Line 166] “we found that zebra finches are sensitive to this color.”
Could the authors please cite a study which examines the opsin sensitivities of zebra finches or other relevant data to back up this statement? If not, that of relevant similar birds or some data showing the spectral sensitivity of the birds?
[Line 169] “The impedances of the loops for”
Please excuse my naivety, but what is a loop in this context? Alternatively, you could simplify this to not mention the loops and just include “The impedances”
[Line 175-176] “This finding suggested that acute recording (recording right after implantation) was not different “
If the amplitude of the VEPs were not statistically different, why is the title of the paper including the word "strong visual evoked potentials"? I think a more accurate title that better reflected the findings might be "Delayed visual evoked potentials in the zebra finch'... with no mention of VEP amplitude.
[Line 195-196] “Interestingly, the frequencies below 30 Hz were significantly stronger in chemically restrained finches.”
What statistical test was used and what were the results?

[Line 200-202]
How were these 3 frequency bands determined? k-means?
[Line 202-204] “This phenomenon could be explained by the high frequency LFPs riding on wide VEPs but not on narrow VEPs generated by a conscious brain”

Could the authors please add a citation which explains why these specific frequency bands are related to 'wide VEPs' or 'narrow VEP's? I do not think this is an adequate explanation for what is causing these different frequency bands. Although the frequency bands may be visually observable as related to the VEPs, it does not explain the underlying neural dynamics and is therefore vague.
In regards to the conclusion section of the article, I think the discussion and conclusion section of the journal article summarizes the findings well – though in some ways it’s unfortunate that this comes late in the manuscript. Hence, I think the authors might be able to model some of the other parts of the manuscript by looking at the discussion section for guidance. Unfortunately, it does not link to a clear prior hypothesis, although it is linked to previous literature on the subject of anesthesia and visual evoked potentials which is appropriate. I do think the literature about visual evoked potentials and anesthesia could be better represented in the introduction and discussion, as well, although this is not an absolute necessity for manuscript acceptance. The conclusions are related to the results presented, which are appropriate.
I include some specific comments about the discussion section here:
[206-216] The rationale is clear here and I wonder why some of this information is not included in the introduction.
[Line 216-217] “Evoked potentials are highly sensitive to fluctuations in physiological parameters, such as peripheral and core body temperature, arterial blood pressure, hematocrit, and others.”
Please add a citation to this sentence. Alternatively, if the information is in the previously mentioned citation... Soghomonyan et al... please make it clear so readers understand where to look for more information.
[Line 223] “Based on previous discussions,”
Consider rewording to indicate which discussions are being referred to.
[Line 233-235] “Combining isoflurane increases flexibility and safety in the protocol as anesthetic depth can be adjusted instantaneously according to the individual’s physiologic response during the procedure.”
Please add a citation to this.
[Line 235] “The midazolam-butorphanol combination demonstrated”
Please add clarification that this was demonstrated by previous research (or the research cited). For instance, “In prior research, the midazolam-butorphanol combination demonstrated…”
[Line 243] “A pilot study”
Is there a citation to this or a way this study (unpublished) can be indicated?
[Line 255] “We think the drugs inhibited whole brain activity.”
This is a reasonable conclusion considering the literature on anesthetics, as this is one of their most observable effects. However, I think a citation is still required. There is one about suppression of high frequencies a few sentences later, so I think it could be brought up first to support the paper’s reasoning.
[Line 272] “provide”
Consider “administer” as an alternative word.
[Line 302-303] “There was little discussion of anesthetic effects on purely visual evoked potentials.”
There is a prior literature of anesthesia on visual evoked potentials, so I do not know why the authors have suggested there isn’t, unless they are specifically referring to these effects in birds. If so, I think that distinction should be made clear.
[Line 304-306] “A similar situation was found again when discussing anesthetic effects in the current study. The proposed effects should be further verified in avian animals and with a well-controlled study in the future.”
This is quite a vague pair of sentences. As a demonstration, if these are read out of context the reader cannot say what the “similar situation” or “proposed effects” are.
[Line 311-314] “On the other hand, our results showed that physical restraint in small animals was not ideal due to the data from physically restrained birds being difficult to acquire as a result of motion artifacts and the attention of the bird. We repeated the procedure in three birds, but only one bird showed VEPs.”
These sentences make clear what the results are, and I think the results could be stated clearly like this throughout the paper.
[Line 316] “distract VEPs”
Please reword this and try to put it in the context of a scientific idea. It does not make a lot of sense.
[Line 316-318] “Although there was minor interference of the received signal with the current anesthetic protocol, the benefits were enough to compensate for those drawbacks.”
I believe the “minor interference” is motion artefacts as mentioned previously, while the “benefits” are not so obvious to me. I think if the sentence and other sentences like this were worded to be more specific then it will be easier to understand.
There is some speculation lacking citations in the discussion, which I have noted. I do not think it would be very difficult to relate the speculation to some literature, or perhaps note that it is speculation.
[Line 253-255] “The threshold of action potentials may be increased by anesthetics, resulting in difficulty with fast responses. However, the amplitudes of the VEPs were not diminished but were enhanced.”
Some wording here is vague, but I think it could be solved by pairing such statements as much as possible with prior literature.

Additional comments

Overall, I think the authors have undertaken the difficult task of measuring visual evoked potentials from a small animal, the zebra finch. This is a notable technical achievement, and hence is an acceptable topic for an original journal article. I think the experiment itself is sound and interesting to read. The technical details are well outlined and the figures are straightforward and appealing to look at.
My first concern is, as this is the first time this type of recording has been done, I think it would be prudent for the authors to include some more background information in the manuscript about the potential usefulness of the zebra finch as a model for studying visual evoked potentials. I think it should be made clear why studying the zebra finch can be informative. This can help with the overall understanding of the experiment’s purpose, help the reader assess whether it has accomplished its goals; and provide ideas how the technique may be applied in future.
My main concern about the manuscript is that the writing is not clear in places due to vague language. At times this interferes with understanding the experiment design, which makes it difficult to judge the study from a reviewer perspective. The unclear language is most noticeable in the abstract and introduction, where I think some rationale behind the experiment is missing. Improving the logical progression of ideas and adding some relevant background information would make a positive difference to the paper, especially in the introduction.
A lesser concern is the authors’s claim that there is not much literature on the visual evoked potential and general anesthesia. Perhaps this is a problem in communication as opposed to a problem in knowledge on the authors’s part, as they discuss some relevant vision papers throughout the manuscript which make good additions to the background information. Nonetheless, I present a few example journal articles which might be relevant to the topic under study for the authors to consider (albeit from human subjects). I think the authors could be more specific by what aspect of the literature they believe is insufficient. It would give more background for the study and also help strengthen their position by highlighting why the zebra finch could be a good model organism for studying VEPs. I picked these papers as they were related to the terms “anesthesia”, “visual evoked potential”, “VEP”, or “ERP”, and were in a reference library I had readily available about anaesthesia. The authors are welcome to decide if this is relevant for their work or not.
Example papers:
Raitta, C., Karhunen, U., Seppälainen, A. M., & Naukkarinen, M. (1979). Changes in the electroretinogram and visual evoked potentials during general anaesthesia. Albrecht von Graefes Archiv für klinische und experimentelle Ophthalmologie, 211(2), 139-144.

Ota, T., Kawai, K., Kamada, K., Kin, T., & Saito, N. (2010). Intraoperative monitoring of cortically recorded visual response for posterior visual pathway. Journal of neurosurgery, 112(2), 285-294.

Bergholz, R., Lehmann, T. N., Fritz, G., & Rüther, K. (2008). Fourier transformed steady-state flash evoked potentials for continuous monitoring of visual pathway function. Documenta Ophthalmologica, 116(3), 217-229.

Mäkelä, K., Hartikainen, K., Rorarius, M., & Jäntti, V. (1996). Suppression of F-VEP during isoflurane-induced EEG suppression. Electroencephalography and Clinical Neurophysiology/Evoked Potentials Section, 100(3), 269-272.

I hope my comments on specific lines of the manuscript help the authors improve it in a future revision. I ask that the authors pay most attention to the comments about clarity and improving the logical progression of ideas. Some further justification on frequency domain analysis is also requested and encouraged, specifically how the frequency domain was divided, unbiased, into the 3 groups. As mentioned earlier, it would also be interesting to know if the authors had tried filtering the movement artefact out. I encourage the authors to investigate filtering techniques if they haven’t already, to try salvage some of the control data that were excluded. If they have tried already, it would be good to know that in the manuscript. If these steps are taken, I think the manuscript could be considered acceptable for publication.

Finally:

I think the graphical figures are reasonable. The code, unfortunately, has some errors. First, at line 13:
sem_LFPB1=std(Zscore_LFPB1,0,2)./(sqrt(Zscore_LFPB1,2)-1);
Error using sqrt
Too many input arguments.
Also, there is an END missing at the end of the function 'function sMap = boxcarSmoothing(map)' (line 53 start of function).
I think these are simple errors to fix, so the authors should have an easy time with it. One final comment is that one citation in the reference list is incomplete and includes only the authors, date, and article title:
[Line 390-391] “Rajan S, Simon M, and Nair D. 2016. Intraoperative Visual Evoked Potentials: There is More to it than Meets the Eye.”
The article in question is from the Journal of Neurology and Neuroscience, and addition to that not being mentioned in the reference, it appears the authors's first names may have been incorrectly listed as surnames. http://www.jneuro.com/neurology-neuroscience/intraoperative-visual-evoked-potentials-there-is-more-to-it-than-meets-the-eye.php?aid=9814

Reviewer 2 ·

Basic reporting

The article by Yu and Hsiao is very well written, has good English, and is well cited. It is very professional and I especially enjoyed how polished the abstract was.

Figures are well made! I like the graphics in Figure 2, and very much like the heatmaps in Figure 4. Figure 3 is presented a little confusingly. I would prefer you use colors for the Bird 2,3,4 (same on left/right side), to better distinguish it from Bird 1. In Figure1, it would be nice to have additional panels showing at least 10 successive traces stacked.

Experimental design

Presents original research, but it is technical in nature (not biological). The experimental design is generally good, but I have some suggestions.

The biggest problems with this paper are a very low n (only one "control" bird and only 3 "experimental" birds).

Also, I would like to see a control where everything is kept exactly the same about the setup, except a black paper 'blind' is lowered right in front of the screen. This is NOT the same as turning the screen off or not delivering stimulus. The traces and averages from this control could be presented as supplemental data to ensure the readers that there is no cross-talk between the visual stimulus itself and the VEPs coming off the electrodes.

Finally, comparing restrained to a cocktail of 3 agents is difficult. It is standard for labs to only use a single agent, instead of 0 or 3. I have some other papers listed below in general, but I would very much like to see the control of just a Benzo agent (diazepam) added in as well.

Validity of the findings

Underlying data was not provided in an open format. Export as CSV or tab-delimited text files in addition to the raw Matlab format so users that do not have Matlab can open.

Underlying data not assessed.

Additional comments

Line 56/57 Somewhat contradicting statements in this sentence.

Other relevant background / discussion about diazepam alone:
https://www.ncbi.nlm.nih.gov/pmc/articles/PMC5532180/
"Prior to ABR testing, birds were sedated with an intramuscular injection of diazepam (Hospria Inc., Lake Forest, IL) at 4 mg/kg body mass, which was further diluted with sterile saline so that the total injection volume remained around 0.2 mL. This is standard in avian studies, and we used a minimal dose to reduce any effects on ABR (Brittan-Powell et al. 2008; Prather 2012). Sedation response to diazepam varied; if subjects did not rapidly become motionless, the bird was retested on a later date and the dose was increased by 0.25 mg/kg (up to 4.75 mg/kg) on subsequent trials. Subjects remained largely motionless for approximately 120 min during data collection."

---

## Round 0.2 · Minor Revisions

Your manuscript has now been seen by the reviewers. You will see from the comments below that some constructive points are worth considering. We therefore invite you to revise and resubmit your manuscript, taking into account these points. Please highlight all changes in the manuscript text file.

·

Basic reporting

I think the manuscript is much improved. It is now easy and straightforward to understand the goals of the study, and the reasons for comparing the towel restrained birds to the sedative-anaesthesia drugged birds are much clearer. The addition of extra background literature is to the manuscript’s benefit, and the reader now has easy access to further reading on the subject of zebra finches and relevant methodology. The added information in the methods also allows other researchers to understand the protocol better, as well as potentially replicate it. I think the manuscript is very close to being ready for acceptance by the journal, but would benefit from a final pass of editing.

I note that there are some grammatical issues that could be adjusted, and may have been inadvertently added during corrections and new additions to the paper. This is the majority of suggestions. I suggest an extra sentence or two in a couple of places for additional literature or for clarification. A few minor changes to the code and figures is also requested. I would also like to draw the authors’ attention to line [270] where a different word replaces the word ‘experiment’. This should definitely be fixed before the paper is published.

As before, I have left line-by-line comments so the authors may benefit from specific feedback. Note that I am referring to the line numbers in the track-changed document with track changes turned on.

Lines 20-22 of the abstract: “Chemical restraint is a common method to minimize motion artifacts in animals when acquiring VEP data, but little is known about its influence on the signal in an avian animal itself.” The “itself” at the end is unnecessary.

For lines 30-31 of the abstract: “The VEPs of control group were recorded after they fully recovered and restrained by towels. The other birds were recorded under anesthesia.” “of control group” to “of the control group”. “and restrained” to “and were restrained”. In addition, I think it is usual to report the Ns within the abstract in brackets, so please do so. E.g. The VEPs of the control group (N = x) were recorded after” and furthermore “The other birds (N = x) were recorded under anesthesia.” You can see an example of this kind of reporting in a similar bird VEP paper in the same journal: https://peerj.com/articles/2217/

In lines 44-45 of the introduction: “ Evoked potentials are voltage fluctuations of nervous systems which triggered by the occurrences of physical or mental events.” Please adjust to “which are triggered” as opposed to “which triggered”.

Line 62: “monkey” to “monkeys”.
Line 66: “watching” to “watched”

Line 69:71: A citation concerning motion artifacts would be of benefit. Otherwise, I think the changes in this section better explain the reasoning for chemical and physical restraint. The monkey examples are a good addition.

Line 79: “that commonly used” to “that is commonly used”
Line 77: “auditory system” to “the auditory system”. “As regard to” to “In regard to”.

Line 79-82: “At least two visual pathways, the tectofugal and thalamofugal pathway, are identified in avian animals (Zeigler et al. 1993). It will be a very intriguing question about how zebra finches use these visual pathways to process color information in their brains.” If the goal is to eventually use the VEPs to study colour vision, I think an extra citations to make the point would be beneficial. I found this paper about ultraviolet vision and mate selection in the zebra finch, and I think it would be relevant, here: Bennett, A. T., Cuthill, I. C., Partridge, J. C., & Maier, E. J. (1996). Ultraviolet vision and mate choice in zebra finches. Nature, 380(6573), 433.

Line 91: “We proposed” to “We propose”
Line 94: “challenging, a study”. To “challenging. A study”

Line 97: This information about the high mortality rate in birds is very interesting. A short additional comment about it could be added as it relates to Brodbelt’s paper. There is a paragraph about it which I quote directly: “Mortality risks in other small animal species were generally higher again than those reported in rabbits. Birds appeared to be at particularly high risk, as were small mammals such as hamsters, chinchillas and mice. It is likely that small body size contributed to these high risks, with all these species having high surface area to volume ratios, again pre‐disposing to hypothermia during anaesthesia (Flecknell 1996b). Additionally, they generally have high metabolic rates and would be prone to perioperative hypoglycaemia until they resumed eating postoperatively (Flecknell 1996a). Because of their small size, their tracheas were less commonly intubated, therefore maintaining a patent airway and adequate ventilation would be more difficult. Only a small number of each species were anaesthetized or sedated and the relative inexperience of veterinary surgeons with these patients was likely to have contributed to the high perioperative mortality risks.”
You could paraphrase to something similar to: It has been hypothesized that the birds’ small size contributes to the higher mortality risk due to higher surface area to volume ratios and predisposing them to complications like hypothermia (Brodbelt et al., 2008).

Lines 102: 104: “which makes it an appropriate for avian species.” To “makes it appropriate for avian species.” (delete the ‘which’).

Line 113: “The goal is for the relatively small animal and difficult anesthesia conditions to highlight the safety and feasibility of the protocol and provide a balanced anesthetic option in small experimental targets undergoing VEP recording projects and test whether the protocol interferes VEP signals.” Here is a suggested rewording of this sentence: “The goal is to highlight the safety and feasibility of this protocol for a relatively small animal and difficult anesthesia conditions, to provide a balanced anesthetic option in small experimental targets undergoing VEP recording projects, and to test whether the protocol interferes VEP signals.”

Line 116: “The current study could provide evidences of VEPs from conscious to anesthetized states in zebra finches.” To “The current study compares VEPs from conscious and anaesthetized states in zebra finches.”

Line 132: In the methods section, I think the addition of new experiments could be more clearly noted. For instance: “Bird 7 was sedated by diazepam. Bird 8 was physically restrained but its visual stimuli were blocked by a block sponge.” An additional line could be added before this: “Additional control experiments were performed on 2 of the 8 animals.”

Line 148: nuclear rotundus (ROT)
175: “surgical” to “the surgical”
180: “2500” is that sampling rate (Hz)? Milliseconds? Seconds?
186: Remove “for humans”.
195: “A towel rolled” to “A towel was rolled”.

Line 200: Please change “their building toolbox and an open source toolbox” to “using the open source toolbox”. Chronux is a third-party toolbox not owned by Mathworks. Furthermore, I do not understand what “building toolbox” is so the sentence makes more sense without it, unless there is a detail I am misunderstanding.

Line 270: A very unfortunate typo here. The sentence should be “restrained experiment again”. Please correct this. It is in the .pdf version as well.
Line 299: “we also interested in” to “we were also interested in”.
Line 299: “influences traditional sedative” to “influences of the traditional sedative”.
Line 304: “For the reason” to “For this reason”.
Line 305: “Fig. S2 show” to “Fig. S2 shows”.
Line 363: “VEP” to “VEPs”.

Line 374-375: Generally speaking, the discussion and results section reads much better with additional references and reasoning. However, this phrasing is vague: “may draw all attention to it” as explaining the reason for the easier-to-observe VEPs. I think a better phrasing concerns an altered signal to noise ratio as discussed briefly here:
Banoub, M., Tetzlaff, J. E., & Schubert, A. (2003). Pharmacologic and Physiologic Influences Affecting Sensory Evoked PotentialsImplications for Perioperative Monitoring. Anesthesiology: The Journal of the American Society of Anesthesiologists, 99(3), 716-737.

Here is a direct quote for reference: “Neuromuscular blocking drugs do not directly influence SSEP, BAEP, or VEP.90 However, they may improve waveform quality by favorably increasing the signal-to-noise ratio through elimination of the electromyography artifact,90 which introduces noise at higher frequencies, especially when EPs are acquired at lower stimulation frequency and higher frequency cutoffs.90”

Even though it is not about either of the drugs in the study, what might be worth mentioning is that the improved ability to observe VEPs might also be due to reduced noise from movement artefacts due to sedation and/or anesthesia, leading to a more favourable signal-to-noise ratio. This is also relevant due to the difficulty in getting good quality data from the control animals, due to movement artifacts. It could be an explanation in addition to suppression of high frequencies from anesthesia. The authors are welcome to find a different citation more suitable to the methods in the paper.

Line 397: “Compared to a previous study” to “In a previous study”
Line 419 to 420: “There was little discussion of anesthetic effects on purely visual evoked potentials in birds.” To “There has been little discussion in the literature of anesthetic effects”…
Line 435: “may cause by” to “may be caused by”
Line 436: “result in s less synchronized” to “results in a less synchronized”.
Line 437-438: “As mentioned in introduction” to “As mentioned previously,”.
Line 438: “contain” to “contains”
Line 439: “not easy for zebra finches” to “not as easy to accomplish in zebra finches”
Line 441: “applying on multi-regional recording” to “application to multi-regional recording”

Experimental design

As mentioned in my previous review, I think the experimental design was acceptable. I think the two additional control recordings as suggested by the other reviewer may not have added significant value, but are interesting nonetheless. I do not understand why a black sponge was used as opposed to a black screen for one of the additional experiments. So I would like to request the authors provide information about what the sponge actually is (including the item name or number? Manufacturer?), because the supplemental figure has an illustration of the sponge as a rectangle, yet there is no information about the sponge itself. It seems quite ambiguous to me what the dimensions of the sponge are. Some more information would be helpful.

That aside, I think it is beyond the scope of this paper to do more experiments, and what is included is sufficient to satisfy the average reader’s curiosity.

Validity of the findings

As mentioned previously, I think the findings are valid. Delayed potentials under anesthesia has been reported in other animals previously and the finding that the combination anesthesia protocol does not suppress visual evoked potentials significantly due to the weakened anesthesia is not an implausible outcome.

The authors have made an effort to save some of the control recordings. However it looks as if the animals were either distracted or the recordings were too contaminated with movement artifacts to see any obvious visual response in the traces. They have provided some of this data in the supplement which I think was a good decision. It is likely an unfortunate consequence of the control method that the animals were moving, hence why the sedation-anesthesia protocol is a potential solution.

Additional comments

I would first like to comment on the figures.

There is a minor typo in Figure 3 for the right-hand column (green light) for Bird 1 vs Bird 3, where Bird 3 is written as “Bird3”.

Supplemental figure 3 (S3) seems to have different y-axis limits for the raw LFP and ICA data (-0.2 to 0.2 and -5 to 5). Is it the case that the ICA data had not yet been z-scored? At the least, I think it would help interpret the figure if the LFPs were presented on the same scale. At this scale, the traces look very different, although it does highlight the problem of having difficulty finding the visual evoked potential. I cannot see them in this figure either. I think it was a good decision to include some of the control animals in the supplement.

Supplemental figure 4 (S4)… Regarding the ICA analysis, there are usually two different approaches to performing ICA. One, as the authors have done, you can do ICA on the data once it has undergone division into trials (epoching). Two, you can perform ICA once on the data before it has been divided into trials. I wonder if the authors have tried this latter approach and if there were any better results? If there is no real difference then it is okay to keep the ICA-corrected approach the same.

I loaded the code in Matlab 2016b, as the authors used the same version.
Regarding the code, I loaded ‘for_peerJ_code.m’ and got a few errors.

Line 33: Undefined function or variable 't'.
I supposed this was time so I added a ‘filler’ variable to the workspace “t = 1:size(mean_zscore_LFPB6,1)” However, the authors may wish to include the actual ‘t’ variable.

Next I noticed the first figure, line 33, would not display. It would when you transpose the data. “plot(t,mean_zscore_LFPB6');hold on” you also have to remove the x-limits to have the data display properly. Why were these [-1 1] x-limits included when they are not allowing the data to be displayed properly?
I also got this error:
Undefined function or variable 'boxcarTemplate2D'.
Error in for_peerJ_code>boxcarSmoothing (line 120)
box = boxcarTemplate2D();
Error in for_peerJ_code (line 74)
B1spectrogram= boxcarSmoothing(nanmean(B1Szscore,3)); %towel-restrained

In regards to Chronux, it would be helpful if the authors could report the version.

In addition, it would help also if the authors could explain their naming convention for each of the bird LFP variable names in the MATLAB comments “%”. For example, in the raw data file there are ‘LFPB1’ and ‘LFPB1_1’, ‘LFPB1_2’, ‘LFPB1_3’. What is the difference between LFPB1 and LFPB1_1? What do the ‘_x’ number labelling signify?

I would like to ask the authors to please have a colleague try to verify the code separately to make sure the code runs without errors. It will save some time later if multiple reviewers are checking in parallel and maybe somewhat inefficiently.

Reviewer 2 ·

Basic reporting

Thank you for addressing my comments. The paper is now ready for publication.

Experimental design

a

Validity of the findings

a

Additional comments

They have done an excellent job addressing my comments, I really appreciate the new experiments. It is ready to get pushed out now. Thanks!

---

## Round 0.3 · accepted · Accept

Thank you for the detailed response letter. We are delighted to accept your manuscript for publication.